# Change Point Detection for Fine-Grained MFR Work Modes with Multi-Head Attention-Based Bi-LSTM Network

**DOI:** 10.3390/s23063326

**Published:** 2023-03-22

**Authors:** Yiying Fang, Qihang Zhai, Ziwei Zhang, Jing Yang

**Affiliations:** 1School of Information and Electronics, Beijing Institute of Technology, Beijing 100081, China; 2School of Cyberspace Science and Technology, Beijing Institute of Technology, Beijing 100081, China

**Keywords:** bi-directional long short-term memory (Bi-LSTM), Change Point Detection (CPD), Multi-Functional Radar (MFR), fine-grained work modes, multi-head attention

## Abstract

Detection of the changes in Multi-Functional Radar (MFR) work modes is a critical situation assessment task for Electronic Support Measure (ESM) systems. There are two major challenges that must be addressed: (i) The received radar pulse stream may contain multiple work mode segments of unknown number and duration, which makes the Change Point Detection (CPD) difficult. (ii) Modern MFRs can produce a variety of parameter-level (fine-grained) work modes with complex and flexible patterns, which are challenging to detect through traditional statistical methods and basic learning models. To address the challenges, a deep learning framework is proposed for fine-grained work mode CPD in this paper. First, the fine-grained MFR work mode model is established. Then, a multi-head attention-based bi-directional long short-term memory network is introduced to abstract high-order relationships between successive pulses. Finally, temporal features are adopted to predict the probability of each pulse being a change point. The framework further improves the label configuration and the loss function of training to mitigate the label sparsity problem effectively. The simulation results showed that compared with existing methods, the proposed framework effectively improves the CPD performance at parameter-level. Moreover, the F1-score was increased by 4.15% under hybrid non-ideal conditions.

## 1. Introduction

Modern Multi-Functional Radars (MFRs) can perform multiple tasks at the same time, including surveillance, tracking and target recognition through the adaptive selection of various complex signal modulation types and parameters. Electronic Support Measure (ESM) systems analyze intercepted radar signals to detect the change points of the work modes to assess threats [1]. These radar signals may contain multiple work mode segments of unknown number and duration. The mode changes are more difficult to detect when the signal features are compromised in non-ideal conditions, such as when pulses are lost or spurious. Therefore, the precise detection of the change point of each work mode is a vital challenge for ESM.

Change Point Detection (CPD) refers to a set of feasible methods for identifying abrupt changes in a time series and can be applied in various fields to extract work mode information [2,3,4,5,6,7]. The objective of this considered task is to detect all points at which a change in trends occurs, which are defined as change points. In particular, CPD has been proven to be effective in radar signal processing. Early studies for radar work mode CPD were typically based on statistical methods, such as probabilistic analysis [8,9], circular binary segmentation algorithms [10], cumulative sum algorithms [11,12,13], and manually-designed feature extraction [14]. Nevertheless, heavy reliance on these hypothetical statistical models and manually-designed features usually results in high complexity and poor performance under non-ideal conditions in real applications. Deep Learning (DL) has provided a promising approach to this problem through automatic learning of complex patterns from large datasets without the requirement for feature and metric engineering [15,16,17]. Most related studies solved the work mode CPD through binary classification or work mode recognition and mode change boundary detection methods. In Refs. [18,19], an Autoencoder was used to remove noise from the original pulse stream before identification. Several studies [20,21,22] have used Convolutional Neural Networks (CNNs) to recognize the modulation type of the Pulse Repetition Interval (PRI).Compared to traditional manual feature extraction, CNNs achieved significant improvements in recognition performance. These CNN-based methods improved MFR system performance and are robust to lost and spurious pulses but have the limitation of requiring inputs of fixed length. To overcome this limitation, in Refs. [23,24,25], Recurrent Neural Networks (RNNs), including their improved form, i.e., The Long Short-Term Memory (LSTM) network, were utilized to classify pulse sequences of different work modes. Moreover, RNNs introduced the concept of timing into network architecture design to achieve better adaptability in time series data analysis. However, due to the vanishing gradient problem, RNNs have a limited memory capacity, which can be problematic for tasks that require the processing of very long sequences. Furthermore, through Ref. [26], attention mechanisms with RNNs made their way into modulation recognition, resulting in an improvement in capturing dependencies at long intervals. These existing DL methods have achieved automatic identification of multiple work modes with different modulation types under non-ideal conditions, which can be understood as detecting change points of work modes at the modulation type level. Considering that within the same modulation type, there may exist consecutive multiple work modes with different modulation parameters in the MFR pulse sequences. These fine-grained work modes exhibit similar trends under the same modulation type, making it difficult for existing methods to extract discernible features. Therefore, it is necessary to further study CPD algorithms at the modulation parameter level to cope with the rapid and flexible work mode changes of advanced MFRs. There are two problems remain to be solved: first, CPD tasks are often treated as binary classification problems in deep learning, which creates label imbalance issues and makes model fitting difficult. Second, the feature extraction module for detecting fine-grained work mode changes needs to be carefully designed.

To address these challenges, this paper propose a processing framework to accommodate fine-grained MFR work modes with flexible modulation types and parameter ranges under non-ideal conditions. The proposed framework consists of three modules: data normalization, label configuration, and a Multi-Head Attention-based Bi-LSTM (MHAB) network. Data normalization ensures that the input radar sequence values are within a unified range. Then label configuration converts sequences with change point labels into weighted change probabilities. The MHAB network identifies all trends with different dynamics, extracts important features and generates the probabilities of change points. The training process is optimized utilizing weighted Binary Cross Entropy (wBCE) as the loss function, which can highlight the minority target labels during training. Extensive simulations with PRI-defined MFR work modes prove the effectiveness and superiority of the proposed method. To our knowledge, this paper is the first to introduce the DL-based CPD methods into fine-grained MFR work mode detection. The main contributions of this paper are as follows:The Bi-LSTM learns the long-term dependencies between past and future, and the Multi-Head Attention (MHA) mechanism helps the model to focus on multiple aspects of the most informative features to reduce the impact of non-ideal observations and other useless features. Thus, the performance of the proposed method on CPD is also improved.This paper design a new label configuration and utilize a weighted binary cross entropy (wBCE) loss function for training, which effectively addresses the problem of sparse change point labels.Simulation results show that the proposed CPD method is superior to other methods for fine-grained work modes, and has robustness under non-ideal conditions.

The remainder of this article is organized as follows: in Section 2, the CPD problem is formulated. In Section 3, the proposed processing framework is presented. The simulations are discussed in Section 4. Finally, Section 5 is the conclusion.

## 2. Problem Formation

### 2.1. Fine-Grained MFR Work Modes

MFR work modes are defined via radar pulses with multiple parameters [27]. Mathematically, it can be represented on two levels from the implementation perspective: modulation type level (modulation-level) and modulation parameter level (parameter-level) [28].

**Definition 1.** 
*A radar **work mode** is described via a dedicated arrangement of a finite number of ordered pulses $P=\left(p_{1},p_{2},\cdots ,p_{T}\right)\in R^{H\times T}$ that serves a certain radar function. A **radar pulse**
$p\in R^{H}$ in the **work mode**
P using H
**mode definition parameters** is represented $p=\left(p_{1},p_{2},\cdots ,p_{H}\right)$.*


**Definition 2.** 
*A **mode definition parameter** describes a certain feature of a radar **work mode**. From radar’s perspective, the **mode definition parameters** can refer to some of the work mode related parameters in a Pulse Descriptive Word (PDW), such as PRI, Radio Frequency(RF), Pulse Width(PW) and intra-pulse modulations.*


**Definition 3.** *A **modulation type** refers to the modulation pattern of a **mode definition parameter**. Typical PRI **modulation types** include constant, jittered and stagger. The corresponding modulation type varies with different **mode definition parameters***.

**Definition 4.** 
*A **mode modulation parameter** refers to a specific parameter defining a **modulation type** (e.g., The mean and variance for a jittered modulation type).*


PRI is the most important parameter in the PDW, due to its inherent temporal characteristics and that it largely reflects the characteristics of the intercepted MFR pulse stream. Therefore, to simplify the analysis, in this paper, only the PRI mode definition parameter is considered. A radar work mode can either be defined on a coarse-grained level through different modulation type [24] or a fine-grained level through its modulation type and parameters [9,27,29]. Consequently, two levels of representation can be defined for each MFR work mode.

The coarse-grained representations of the modulation-level work modes underlie the flexibility of MFRs to adapt their transmission modulation types. Specifically, pulse sequences with the same PRI modulation types but different modulation parameters are regarded as the same work mode at the coarse-grained analysis level.

The fine-grained representation of a parameter-level work mode can be understood as an unrestricted choice of the modulation parameters regulated by the relevant modulation type. In particular, since the MFR can select parameter values in the corresponding space, each resulting combination can be a parameter-level work mode. Taking Figure 1 as an example, the PRI sequence includes the four work modes A–D, which are defined as having the same modulation type as jittered but with different parameters in the $\left\{mean,variance\right\}$ space.

#### 2.1.1. Basic PRI Models under Ideal Conditions

The PRI data are inherently timing data and have more complicated temporal correlations than other mode definition parameters, such as the RF. To enhance the applicability of the method, in [30] four statistical PRI models were formulated, including jittered, sliding, stagger and periodic, to fit all modulation types. Other types of modulation can be extrapolated from variations of these basic modulation types. For instance, constant modulation with noise can be considered as jittered modulation with small variance, while dwell-and-switch modulation can be treated as multiple consecutive constant modulations.

#### 2.1.2. Non-Ideal Observation of PRI Sequence

As real electromagnetic environments are very complex, the received signals are inevitably non-ideal, and the interference usually takes the form of measurement noise, lost or spurious pulses. In practical applications, instead of directly observing the PRI values, the Time Of Arrival (TOA) of each pulse is recorded by the ESM receiver. Many factors affect the measurement accuracy of TOA, the most dominant of which is the noise in the receiver circuit. In this study, to simulate measurement noise, Gaussian White Noise (GWN) is added to the original TOA pulse sequence. The crowded electromagnetic environments and reception errors are the main causes of lost and spurious pulses.

$x=\left\{x_{1},x_{2},\ldots ,x_{L}\right\}$ denotes an original PRI sequence of length *L*. The TOA of the *t*-th pulse can be described as ${\mathrm{TOA}}_{t+1}={\mathrm{TOA}}_{t}+x_{t},1\leq t\leq L$. A non-ideal PRI sequence where *j* pulses are lost from the *i*-th pulse can be expressed as [26]:(1)$$x_{t}^{\prime }=\left\{\begin{array}{cc} x_{t}, & t=1,2,\ldots ,i-1\\ x_{i}+\ldots +x_{j}, & t=i\\ x_{t+j+2}, & t=i+1,\ldots ,L-j-1 \end{array}\right.$$

For spurious pulses, a non-ideal PRI sequence with j pulses train insertion after the *i*-th pulse can be expressed as:(2)$$x_{t}^{\prime \prime }=\left\{\begin{array}{cc} x_{t}, & t=1,2,\ldots ,i-1\\ s_{t-i}, & t=i,\ldots ,i+j\\ x_{t-j}, & t=i+j+1,\ldots ,L+j-1 \end{array}\right.$$
where $s_{0}$ is the difference between the original *i*-th pulse and the first spurious pulse; $s_{j}$ is the difference between the last spurious pulse and the original (i+1)-th pulse; and $s_{1}$ to $s_{j-1}$ is the differences between the spurious pulses themselves. An example of lost and spurious pulse conditions is shown in Figure 2.

### 2.2. CPD for PRI Sequence

#### 2.2.1. Change Point Definition

Assuming that the PRI values of a received MFR pulse sequence are denoted using $x_{t}$, where t is the discrete time index. In a certain work mode, ${\left\{x_{t}\right\}}_{t\geq 1}$ follows a certain probability density function $\mathrm{PDF}\left(x_{t}\right)$, with parameter θ.At an unknown time $t_{0}$, the MFR changes the radar work modes with the corresponding parameter changing from $\theta_{0}$ to $\theta_{1}$ after $t_{0}$. The pre-change and post-change distributions $\mathrm{PDF}\left(x_{t}\right)$ are represented respectively as:(3)$$PDF_{\theta }\left(x_{t}\right)=\left\{\begin{array}{c} PDF_{\theta_{0}}\left(x_{t}\right),\;\mathrm{if}\;t<t_{0}\\ PDF_{\theta_{1}}\left(x_{t}\right),\;\mathrm{if}\;t\geq t_{0} \end{array}\right.$$

In this paper, the change point is defined as the first point of the $\theta_{1}$ distribution, denoted by the time index $t_{0}$ in Figure 3.

As there are often multiple change points in a sequence, the set of PRI sequence with *K* change points is formulated as $T=\left\{T_{1},T_{2},\ldots ,T_{K}\right\}$.

#### 2.2.2. Fine-Grained MFR Work Mode CPD Task

In fine-grained work mode, MFRs can emit specified pulse sequences with different PRI modulation types and corresponding modulation parameters. The CPD task’s objective is to accurately identify these work mode change boundaries in the PRI sequence.

An input PRI sequence x of length L input is denoted as $x=\left(x_{1},x_{2},\cdots ,x_{T}\right)\in X$, where ${\left\{x_{t}\right\}}_{1\leq t\leq L}$ is the PRI value of the *t*-th pulse. The corresponding true label is denoted as $y=\left(y_{1},y_{2},\cdots ,y_{T}\right)\in Y$, which expresses the probabilities that the corresponding pulses are the change points. Where X and Y denote the input and the target spaces, respectively. The main task is to learn the model function f:X→Y, from a training dataset $D=\left\{\left(x_{i},y_{i}\right),1\leq i\leq N\right\}$. For each training sample $\left(x_{i},y_{i}\right)$, $x_{i}\in X$ is the input PRI sequence, and $y_{i}\in Y$ is the corresponding output associated with $x_{i}$. Given a test sample x∈X, the trained function f(·) will output $\hat{y}=f\left(x\right)$ as the CPD result, $\hat{y_{t}}$ is the predicted probability that the point corresponding to index t is a change point. As mentioned above, CPD is generally regarded in DL as a problem of binary classification, that is, the probability of judging whether the current point is a change point based on existing information. For this task, a special loss function and a unique neural network structure parameterized using a vector of weights need to be designed.

## 3. The Proposed Approach

### 3.1. The Proposed Framework for CPD

In this paper, a processing framework using a MHAB network for CPD is proposed. As shown in Figure 4, the framework consists of three modules. First, the values of the input PRI sequence are normalized into a unified range. Then, the label configuration module converts the corresponding change points set into a sequence of weighted change probabilities. Next, the MHAB network extracts features and finally generates the target probability sequence. In addition, a wBCE loss function is adopted to improve the fitting of the proposed network. The details of each of these processing steps are described in the following sections.

### 3.2. Data Normalization

PRI sequences are usually represented via different modulation types and parameters with different value ranges. Normalization generally limits data to a specific range, such as [0, 1], thus eliminating the impact of data dimensions on modeling. Data normalization can result in accelerated training speeds and improved network convergence. In this paper, the input sequences are normalized using maximum-minimum normalization according to the following equation:(4)$$x^{{}^{\prime }}=\frac{x-min}{max-min}$$
where min and max denote the minimum and maximum values of the sample sequence, respectively.

### 3.3. Label Configuration

As mentioned above, CPD is a problem of determining whether a given time step is a point of change based on existing information. In DL, these types of problems are typically approached through binary classification. However, the number of change points in a time series is often small compared to the total number of samples; for example, there may be 1 to 4 change points in a sequence with a length of 800 samples. This leads to a highly imbalanced classification problem that can be difficult to model with deep networks. Furthermore, pulse merging and imperfect receiver circuits are factors that often affect real-world radar data, which can cause confusion at mode changes and make it difficult to accurately identify change points. This is particularly true in cases of slow-trend changes. To balance the proportion of samples in the training set and improve the model’s performance, in this paper a method of linear smoothing of the training labels at change points is proposed to improve detection accuracy. As shown in Figure 5, if the change point’s position is $T_{i}$ and a smoothing margin *M* is used, then the label of each sample is:(5)$$Y_{t}=\left\{\begin{array}{c} 1-\frac{\left|T_{i}-t\right|}{M},\left|T_{i}-t\right|\leq M\\ 0,\left|T_{i}-t\right|>M \end{array},t=1,2,\ldots ,L;i=1,2,\ldots ,k\right.$$

In this new label configuration, model parameters are optimized to solve a regression problem.

### 3.4. MHAB Networks

#### 3.4.1. Feature Extraction Module

The normalized PRI sequence is fed into the two Bi-LSTM layers with rectified linear unit activations. This part regards the entire PRI sequence as a whole and extracts the temporal features for CPD task. The Bi-LSTM iterates over pulse sequences x from opposite directions of forward (superscript fw ) and backward (superscript bw), generating hidden state vectors $H^{fw}=\left(h_{1}^{fw},h_{2}^{fw},\ldots ,h_{L}^{fw}\right)$ and $H^{\mathrm{bw}}=\left(h_{1}^{bw},h_{2}^{bw},\ldots ,h_{L}^{bw}\right)$, respectively, where:(6)$$h_{t}^{fw}=\mathrm{LSTM}\left(x_{t},h_{t-1}^{fw}\right),t=1,2,\ldots ,L$$
(7)$$h_{t}^{bw}=\mathrm{LSTM}\left(x_{t},h_{t-1}^{bw}\right),t=L,L-1,\ldots ,1$$

Here $x_{t}$ is the input pulse of the LSTM at time index t, and $h_{t}$ is the hidden state. LSTM [31,32] denotes the LSTM cell function, which is implemented through the following functions:(8)$$f_{t}=\sigma \left(W_{f}\cdot \left[h_{t-1},x_{t}\right]+b_{f}\right)$$
(9)$$i_{t}=\sigma \left(W_{i}\cdot \left[h_{t-1},x_{t}\right]+b_{i}\right)$$
(10)$${\tilde{C}}_{t}=\tanh\left(W_{C}\cdot \left[h_{t-1},x_{t}\right]+b_{C}\right)$$
(11)$$C_{t}=f_{t}⊙C_{t-1}+i_{t}⊙{\tilde{C}}_{t}$$
(12)$$o_{t}=\sigma \left(W_{o}\cdot \left[h_{t-1},x_{t}\right]+b_{a}\right)$$
(13)$$h_{t}=o_{t}⊙\tanh\left(W_{t}\right)$$
where $f_{t},i_{t},o_{t}$ variables represent the forget gate vector, input gate vector and output gate vector; ${\tilde{C}}_{t}$ is the new candidate value for the cell state, $C_{t}$ and $C_{t-1}$ are the new and the old cell state; W and b with corresponding gate subscripts denote the learnable weights and the corresponding biases; σ and ⊙ are the sigmoid function and the Hadamard product, respectively. Then the Bi-LSTM output’s hidden state $h_{t}$ concatenates the $h_{t}^{fw}$ and $h_{t}^{bw}$, namely $h_{t}=\left[h_{t}^{fw},h_{t}^{bw}\right]$, which can effectively capture the contextual information and dependencies in the input pulse $x_{t}$.

#### 3.4.2. Multi-Head Attention Module

Lost and spurious pulses can cause incorrect PRI values that interfere with detection, yet these values also contain some original information. However, the LSTM lacks the ability to focus on certain areas adaptively, potentially leading to redundancy or lost information during learning process. To address this issue, the MHA mechanism is employed to enable the LSTM to concentrate on the critical feature vectors. This approach enhances the model’s ability to capture the most relevant information from the input data, leading to more accurate and efficient detection outcomes.

The attention mechanism is simulating the data retrieval process in the data management system. To retrieve data, a query needs to be addressed to the data management system. If the query matches a key, the value associated with the key will be retrieved. Equation (Equation 14) shows the formulation process of queries *Q*, keys *K*, and values *V*.
(14)$$\left(Q,K,V\right)=\left(W^{Q},W^{K},W^{V}\right)\cdot H$$
where $W^{Q},W^{K},W^{V}$ are trainable weight matrices. To retrieve the most relevant part of the values *V*, the scaled dot-product method uses to calculate attention [33]:(15)$$\mathrm{Attention}\left(Q,K,V\right)=\mathrm{softmax}\left(\frac{QK^{T}}{\sqrt{d_{k}}}\right)V$$
where $d_{k}$ is the model dimension; $QK^{T}$ is used to calculate the self-attention score which is divided each by $\sqrt{d_{k}}$; and a softmax function is employed to obtain the weights of the values.

As shown in Figure 6, the MHA mechanism is a projection of *Q*, *K*, and *V* through different linear transformations for h times, which corresponds to perform the attention function on different projected versions in parallel. These are concatenated and once again projected, resulting in the final values. This structure allows the model to focus on different aspects than the single self-attention.The equations are as follows.
(16)$$a_{t}=\mathrm{Concat}\left({\mathrm{head}}_{1},\ldots ,\;\mathrm{head}{\;}_{h}\right)W^{o}$$
(17)$$\;\mathrm{head}{\;}_{i}=\mathrm{Attetion}\left(QW_{i}^{Q},KW_{i}^{K},VW_{i}^{V}\right),1\leq i\leq h$$
where $W^{0}$ are the trainable weight matrices of the final projection.

#### 3.4.3. Classifier

Finally, the classifier outputs the probability of change point at time t from the Fully Connected (FC) layer through the sigmoid activation function σ. Given the features $a_{t}$ extracted by the MHA module at time step *t*, the classifier provides the predicted output ${\hat{y}}_{t}\in \left[0,1\right]$, which is the probability of input sample $x_{t}$ being a change point.
(18)$${\hat{y}}_{t}=\mathrm{FC}\left(a_{t}\right)=\sigma \left(W_{\mathrm{FC}}\cdot a_{t}+b_{\mathrm{FC}}\right)$$
where $W_{\mathrm{FC}}$ and $b_{\mathrm{FC}}$ are the weight and bias of the FC layer, respectively.

Another solution to the problem of unbalanced data that we adopted is to reweight the losses to give higher weight to the error relative to the change points. Thus, the wBCE loss function is obtained. By adjusting the weight of the loss function, the model can be forced to pay more attention to the minority target label during training, which improves the accuracy of detecting minority targets.
(19)$$\;\mathrm{loss}\;=-\frac{1}{L}\sum_{t=1}^{L}w_{c}\cdot y_{t}\cdot \log\left({\hat{y}}_{t}\right)+\left(1-y_{t}\right)\cdot \log\left(1-{\hat{y}}_{t}\right)$$

In this paper, $w_{c}$ is set to $w_{c}=\left(N_{N}/N_{\mathrm{CP}}\right)$, with $N_{N}$ and $N_{CP}$ being the total numbers of normal points and change points in the sequence, respectively. The use of this loss function for updating results in better model performance.

## 4. Simulations and Analysis

In this section, four non-ideal scenes are simulated to evaluate the ability of the proposed framework for fine-grained MFR work mode CPD with different PRI sequences.

### 4.1. Simulations Design

#### 4.1.1. Dataset Description

In this paper, the MHAB network is evaluated on a simulated PRI sequence set with change points. Four PRI modulation types were adopted in the sample sequences including jittered, periodic, sliding and stagger. The simulation data were generated randomly based on the modulation types and the modulation parameter values were generated uniformly from specific ranges. The corresponding value ranges of the modulation types and parameters during training and testing are listed in Table 1. An L length PRI sequence with K change points was utilized as a sample, where L = 800, K = U(1, 4). The interval between adjacent change points was greater than 100.

In order to evaluate the robustness of the proposed method in non-ideal conditions, four scenarios were simulated, including measurement noise, lost pulses, spurious pulses and the hybrid scenes combinations of these three. The three basic non-ideal condition scenarios are denoted as Measurement Noise Only (MNO), Lost Pulse Only (LPO), and Spurious Pulse Only (SPO). In the MNO scenario, GWN was introduced to the TOA observation at seven noise levels with a variance σ ranging from 0 µs to 3 µs with a step of 0.5 µs. Both LPO and SPO involved pulse sequences with a proportion of lost or spurious pulses at five levels ranging from 0 to 20% in 5% steps. The hybrid scenario combined the three basic non-ideal conditions, the parameter setup as shown in Table 2. Training and test sets were generated to verify the robustness of the model under these non-ideal scenarios. Taking lost pulse only scenarios as an example, 4000 samples per lost pulse ratio level were generated, i.e., 20,000 in total for five levels. The samples are divided into training and test sets with a proportion of 75% and 25%, respectively. Then the model was trained using the 15,000 samples in training sets together and separately tested with 1000 samples at each level. The dataset settings were the same for other scenarios.

Simulations were also conducted to verify the extent to which the model trained on MNO was affected by the modulation types and the number of change points.

#### 4.1.2. Evaluation Metrics

The ability to detect change points within a specific margin of error is an important aspect of evaluating CPD methods [2]. F1-score can represent the detection accuracy within a certain error range. Therefore, in this study, the F1-score is used to evaluate the performance of the models. A change point estimation was considered to be a True Positive (TP) if it fell within an uncertain detection margin M of the ground truth change point. In cases where multiple change point estimates fell within the error margin of the ground truth change point, only the closest estimate was considered a true positive, and the rest were considered False Positives (FP). Ground truth change points without any estimates within the specified error margin were considered False Negatives (FN). In this simulation, the detection margin M and the target region of label configuration were set to 5.

#### 4.1.3. Simulation Implementation

The Adam optimizer [34] was adopted with a learning rate $l_{r}$ of 0.00005 ($5\times 10^{-5}$). For the network layers, the network weights were updated using the following rule:(20)$$W=W-l_{r}\frac{m}{\sqrt{\nu }+\varepsilon }$$

The number of hidden layer nodes $N_{h}$ of the network was set to 256, and the size of the input batch was 128. In addition, in order to mitigate overfitting, the dropout strategy was employed for the Bi-LSTM layer. After multiple rounds of simulations, when the dropout rate was 0.25, the model achieved a better performance. Following [33], the number of attention heads h was set to 8. Early stopping was been employed in the simulations to avoid overfitting during training. The programing environment of the proposed model was Python 3.7.0 and the deep learning library Pytorch 2.6.0 was used.

### 4.2. Validation of Basic CPD Performance

To demonstrate the effectiveness in CPD, this section compared the performance of the proposed method with three methods: CNN [2,18,19], CNN-LSTM [2] and Bi-LSTM [20,21,31] as baselines. Ref. [18] extended PRI modulation recognition to PRI CPD on the basis of CNN. The hybrid CNN-LSTM model [2] combined CNN and LSTM which extracting complex features effectively with spatial and temporal information. With its inspiration, this paper applied CNN-LSTM from anomaly detection in our MFR problem. The Bi-LSTM for radar pulse sequence analysis [31] and a version of Bi-LSTM involving weighted BCE loss function (Bi-LSTMw) [35] were both adopted as baselines.

In order to verify the basic performance of CPD for fine-grained MFR work modes, 1000 test samples with different parameter were generated at each modulation type. The model were trained by the MNO scenario at all variance levels and tested at the σ = 1.5 µs level. Due to the sparsity of change points in the samples, the baseline methods could hardly reach convergence in our simulation scenario with the normal binary labels, so label configuration was added to all baseline methods. In the methods using wBCE loss function, predictions were obtained by retaining the peaks in the output probability distribution with threshold values greater than 0.8, while for the other methods, the threshold was set to 0.3.

The different methods’ CPD performance on PRI sequences of different modulation types is presented in Table 3. It can be seen from Table 3 that the MHAB achieved the best effect under all the modulation types. Because the LSTM can learn long time sequence information, it showed a significant improvement compared with CNN. CNN-LSTM combines local spatial information and long temporal information, thus reducing the error rate. The Bi-LSTM allowed the model to learn the dependencies between the past and future of the data to better capture two-directional features, so its detection effect was also improved. The weighted BCE loss function constrained the acceleration of the network’s convergence and reduced the possibility of the output being drowned in noise, while the MHA mechanism further enhanced this effect. Since the constraint of wBCE loss function made the probabilities obtained by the network closer to 0 and 1, it can be seen from Figure 7 and Figure 8e that the predicted values of MHAB and Bi-LSTMw at the change point were also the highest, reducing the detection error caused by the peak threshold setting.

Compared with the randomly jittered PRI, the modulation characteristics of more regular sequences were more easily learned by the network. Therefore, the detection effect for periodic, sliding and stagger PRI sequences was better.

### 4.3. Performance under Non-Ideal Conditions

Electromagnetic environments with highly non-ideal conditions often result in contaminated pulse sequences. Effective CPD methods should be robust enough to correctly recognize change points in corrupted pulse sequences. The models’ performance was therefore evaluated in four non-ideal scenarios, including MNO, LPO, SPO and hybrid scenario at 7, 5, 5 and 4 levels of interference intensity, respectively. In each scenario, the model was trained using 3000 samples per non-ideal condition intensity level together. And 1000 samples were tested on each level. The F1-score was used to evaluate the performance of the different methods, and the corresponding simulation results are shown in Figure 8.

The MHAB significantly outperformed the other baseline methods in all non-ideal conditions. Figure 8a shows that the performance of all methods was almost unaffected by the increase in measurement noise variance. Figure 8b,c show that the total F1-score of MHAB was almost unaffected by the proportion of lost and spurious pulses, but the total F1-score of CNN, CNN-LSTM and Bi-LSTM decreases as the conditions become worse. In Figure 8d, as the hybrid conditions became more complex, the total F1-score of all methods decreased, inevitably. But the proposed method proved to be more stable. Because it lacks the ability to capture timing relationships, CNN exhibited the worst performance among all methods. Compared with CNN-LSTM and Bi-LSTM, Bi-LSTMw achieved better performance, thanks to the use of wBCE in training. However, the effective improvement of Bi-LSTMw due to the utilization of wBCE loss became less obvious as the percentage of spurious pulses increased. This was primarily because the original sequence features were not effectively extracted, owing to the overwhelming presence of noise. To compensate for this shortcoming, the MHA mechanism was adopted in the proposed method. As shown in Figure 8e, the stagger PRI sequence exhibited five work modes and corresponding four change points in the hybrid 4 scenario. However, the transition between the fourth and fifth work modes was not clearly visible in the noise. Including Bi-LSTMw, none of the four baselines were able to detect this change point. With the help of MHA, MHAB was able to focus on this transition well. Although hybrid scenes posed a higher challenge to feature extraction, the proposed method still achieved a 93.83% CPD accuracy with 15% spurious pulse, 15% lost pulse and 3 µs measurement noise.

### 4.4. Comparison of Different Framework Structures and Experimental Settings

#### 4.4.1. Influence of Label Configuration

This section examined the influence of the designed label configuration. F1-score was used to present the results. In Section 3.3, the smoothing label sequence is designed to address the pulse confusion problem that arises at change points in practical situations. It also solves the difficulty of model fitting caused by sparse samples to obtain better performance.

Figure 9a compared the performance of the original binary label and designed label. With different modulation types, the F1-score for designed label was considerably higher than original label corresponding to all the four types. The maximum gap was 8.08%. Summarizing, the superiority of label configuration in this study was verified.

#### 4.4.2. Influence of WBCE Loss Function

As discussed in Section 3.4.3, another solution to the problem of unbalanced data that we adopted is wBCE loss function. This section compared the performance of the MHAB network trained by binary cross entropy (BCE) loss function and wBCE loss function. The result was presented by F1-score.

The periodic and sliding modulation types exhibit obvious trends, making the ordinary BCE loss function sufficient for the model to learn these features. However, the weak features of random jittered and stagger modulation in MFR sequences submerge the noise, which pose a great challenge to BCE loss function. As shown in Figure 9b, the effect enhancement of the network using wBCE loss function was more significant for jittered and stagger types. The F1-scores of the former methods increased from 87.83% to 97.34% and from 91.13% to 99.72%.

#### 4.4.3. Influence of Change Point Numbers

The variation of the number of change points causes various detection difficulties. Therefore, in this simulation the network was trained uniformly through the introduction of K = 1 to 4 change points in MNO scenario and tested at the σ = 1.5 µs level. 1000 test samples were generated at each level. The FN and F1-score were measured respectively under different values of K. The corresponding results are shown in Figure 10. Intuitively, with the increase of the change point number, the number of false positives increased. On the whole, the FN was more sensitive to K than the total F1-score, which means that FN should be monitored more closely in CPD tasks. Compared with the other baseline methods, the FN increase rate as K increased was reduced. This showed that the proposed model can adapt to more complex samples, and the FN for samples containing four change points was only 59 points out of 4000 total predicted points.

## 5. Conclusions

In this paper, an MFR work modes CPD framework using the MHAB network is proposed. In this framework, the Bi-LSTM learns the dependencies of contextual data in the pulse stream, while the attention mechanism emphasizes the informative features and ignores the noisy parts. Meanwhile, a new label configuration is adapted and an improved wBCE loss function is utilized for training, which solves the problem of change point label sparsity effectively. The simulation results demonstrate that in each modulation type, the proposed framework effectively extracts the temporal features between pulses of different modulation parameters and can be used to detect changes in work modes in different non-ideal environments. In future work, we will perform the detection using real data and conduct this research in unsupervised methods.

## Figures and Tables

**Figure 1 sensors-23-03326-f001:**
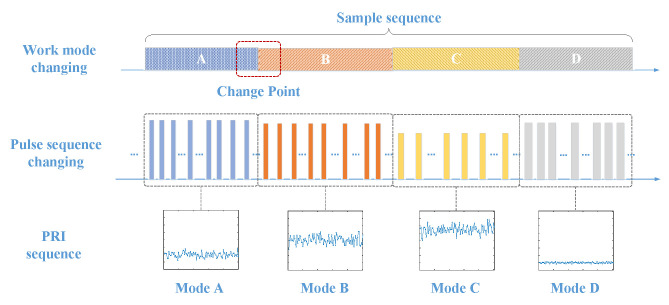
Illustration of PRI sequence with fine-grained MFR work modes conversion.

**Figure 2 sensors-23-03326-f002:**
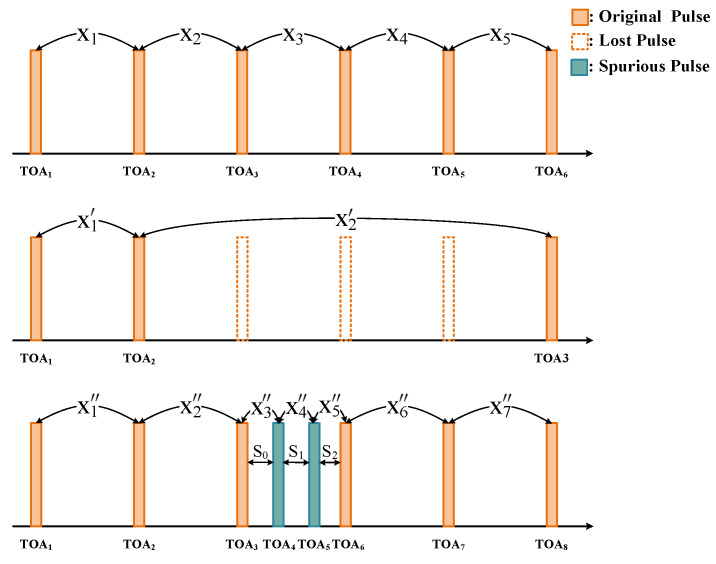
Example of lost and spurious pulse conditions.

**Figure 3 sensors-23-03326-f003:**
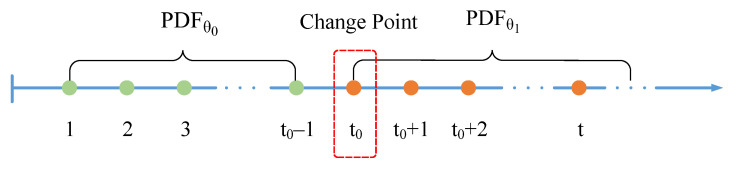
A single change point model.

**Figure 4 sensors-23-03326-f004:**
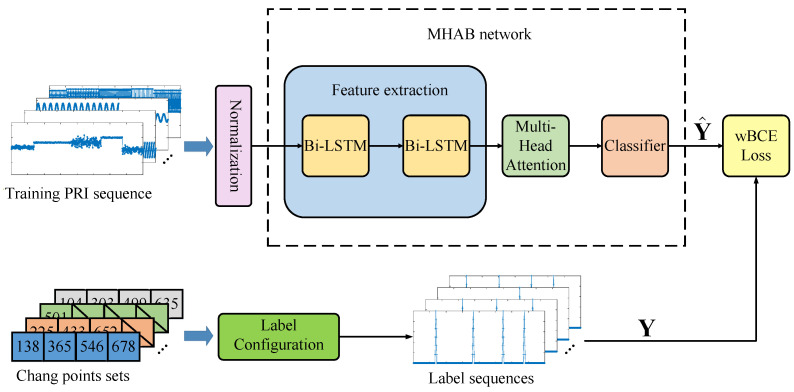
Flow diagram of the proposed MHAB-based CPD framework.

**Figure 5 sensors-23-03326-f005:**
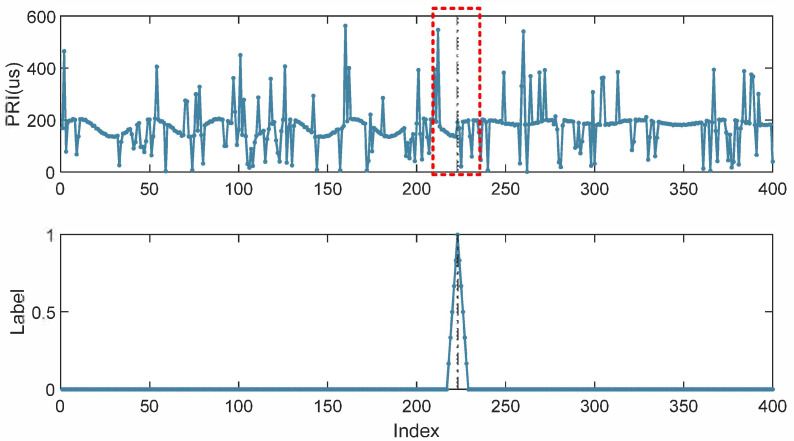
Example of the change point in a noisy periodic PRI sequence (**first row**). Both the dynamic trend and noise makes it difficult to establish the exact location of the change point, which can be identified within the red dashed box region. The corresponding label designed as a linear centered smoothing margin in the change point (**second row**).

**Figure 6 sensors-23-03326-f006:**
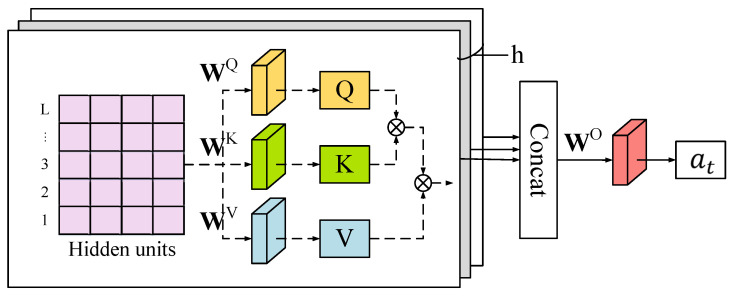
The framework of the MHA mechanism.

**Figure 7 sensors-23-03326-f007:**
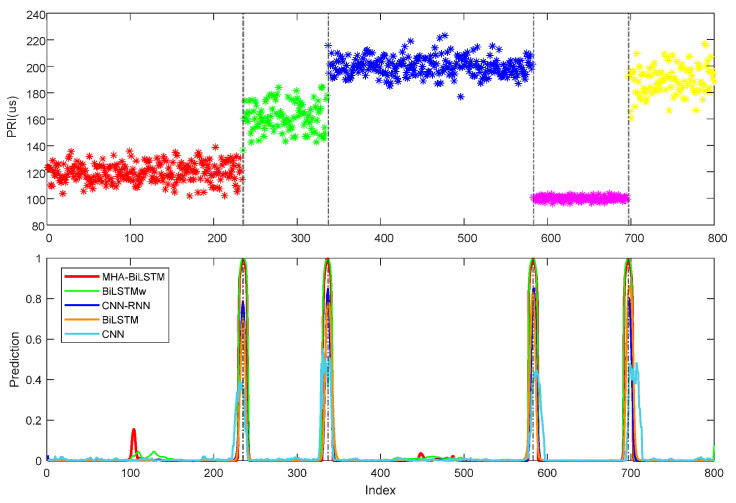
Example of predicted label sequences on jittered PRI, including 4 change points (gray dashed). The horizontal axis is the PRI index, and the vertical is the probability that the point is a change point.

**Figure 8 sensors-23-03326-f008:**
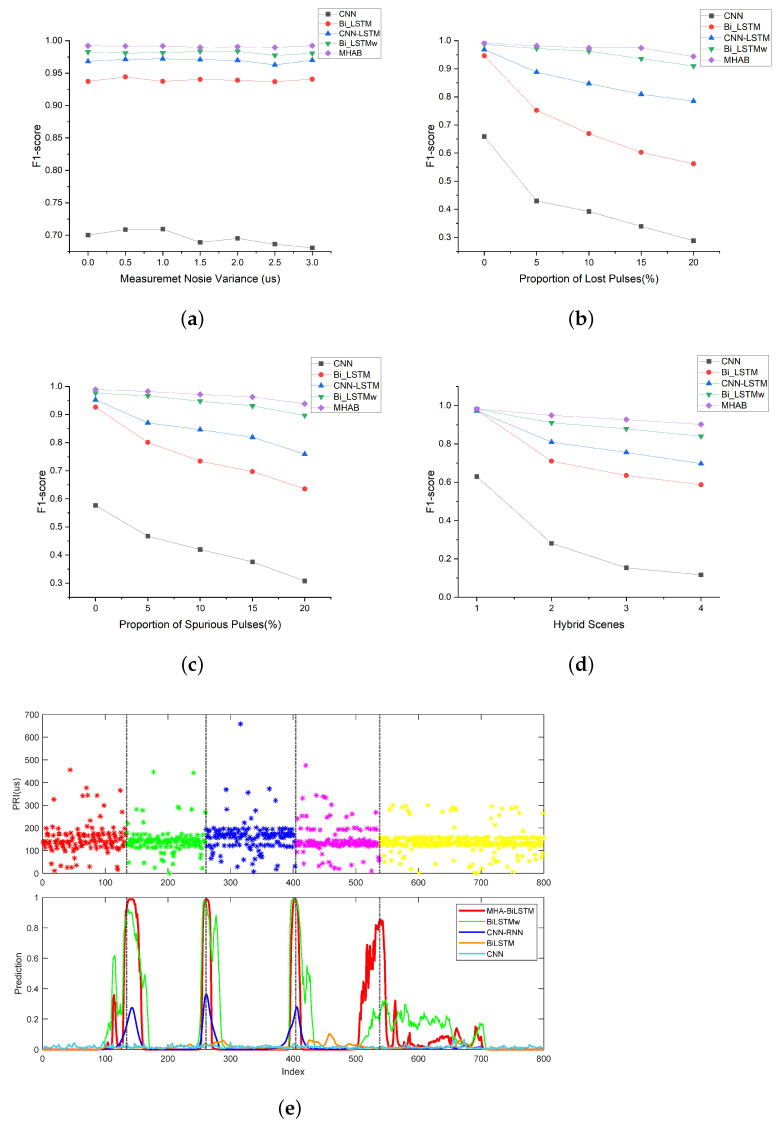
CPD performance under non-ideal conditions. (**a**) Under measurement noise conditions; (**b**) Under lost pulses conditions; (**c**) Under spurious pulses conditions; (**d**) Under hybrid non-ideal conditions; (**e**) A predicted result of a stagger PRI sequence in hybrid scene 4.

**Figure 9 sensors-23-03326-f009:**
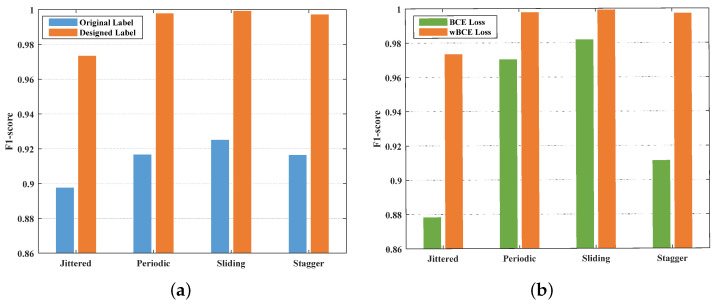
Influence of label configuration and wBCE loss function. (**a**) Performance comparison of original label and designed label; (**b**) Performance comparison of BCE loss and wBCE loss.

**Figure 10 sensors-23-03326-f010:**
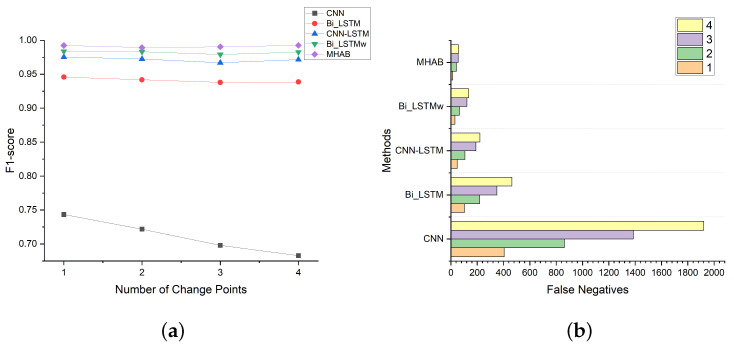
The effectiveness of change point numbers; (**a**) Result presented by F1-score; (**b**) Result presented by FN.

**Table 1 sensors-23-03326-t001:** Value ranges of modulation types and parameters.

Types	Parameters	Ranges
Jittered	Mean of jittered value	U(100 µs, 200 µs) ^1^
	Variance of jittered values	[1 µs, 4 µs, 7 µs, 10 µs]
Periodic	Center value	U(100 µs, 200 µs)
	Modulation amplitude	U(10%, 20%)
	Sampling frequency	U(2 fc, 8 fc)
	Center frequency	50 Hz
Sliding	Initial value	U(10 µs, 30 µs)
	Rate	U(2, 6)
	Number of sliding steps	U(10, 30)
Stagger	Range of stagger value	U(100 µs, 200 µs)
	Number of stagger steps	U(3, 10)

^1^ U(·) represents the uniform distribution.

**Table 2 sensors-23-03326-t002:** Parameter setup of hybrid scenes.

Scene	Measurement Noise (µs)	Lost Pulse (%)	Lost Pulse (%)
1	0	0	0
2	1	5	5
3	2	10	10
4	3	15	15

**Table 3 sensors-23-03326-t003:** Effectiveness of four modulation types.

Method	Jittered			Periodic			Sliding			Stagger		
	TP↑	FN↓	F1↑ ^1^	TP↑	FN↓	F1↑	TP↑	FN↓	F1↑	TP↑	FN↓	F1↑
CNN	1236	1234	0.6670	1403	1089	0.7202	1154	1299	0.6392	1560	940	0.7641
Bi-LSTM	2211	259	0.9447	2036	457	0.8991	2318	135	0.9717	2224	276	0.9416
CNN-LSTM	2221	249	0.9469	2317	175	0.9636	2315	138	0.9711	2396	104	0.9788
Bi-LSTMw	2266	204	0.9569	2428	84	0.9870	2422	31	0.9936	2448	52	0.9895
MHAB	**2342**	**128**	**0.9734**	**2481**	**11**	**0.9978**	**2449**	**4**	**0.9992**	**2486**	**14**	**0.9972**

^1^ F1 means F1-score. ↓ means that smaller is better for this indicator; otherwise, it’s ↑. Bold values represented the best.

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
