# Peer review of "Change Point Detection for Fine-Grained MFR Work Modes with Multi-Head Attention-Based Bi-LSTM Network"

_sensors, 2023, doi:10.3390/s23063326_

Round 1

Author Response

Dear reviewer:

Thank you for your careful reading and the comments concerning our manuscript. Those comments are valuable and very helpful. We have read through comments carefully and have made corrections. Based on your comments, we uploaded the file of the revised manuscript. Revisions in the text are shown using blue highlight for additions, and strikethrough font for deletions. The responses to the reviewer's comments are presented following.

Point 1: The whole paper needs a thorough review for language errors, sentence construction, word use and grammar.

Response 1:

We apologize for the confusion caused by language errors, sentence construction, word use and grammar. We have checked the whole paper repeatedly and corrected them and we marked these parts in blue in the revised paper. The manuscript has been thoroughly revised and edited by a native speaker. We hope it can meet the journal's standard. Thanks so much for your useful comments.

Point 2: In the ‘Methods’, what is the innovation of these methods? It seems to be a simple combination of the existing LSTM techniques. It needs further innovations to support the research work.

Response 2:

We apologize for making confusion about our innovative points. We have separated our contribution to the LSTM’s discussion in the Introduction.

In this paper, we solve the problem of MFR work mode change point detection at parameter-level. The existing LSTM techniques have two problems adapting to our task:

1) The methods of solving CPD through binary classification may cause data imbalance and are difficult to converge;

2) These fine-grained work modes exhibit similar trends under the same modulation type, making it difficult for existing methods to extract discernible features. Therefore, the feature extraction module needs to be carefully designed to detect the parameter changes fine-grained work modes.

To address these challenges, this paper proposed a processing framework to solve these problems as shown in Figure 4 with label configuration, MHAB network and wBCE loss function. The proposed MHAB network improves feature extraction by leveraging the Bi-LSTM's ability to learn long-term dependencies between past and future and the MHA mechanism's ability to focus on informative features. Thus, the proposed MHAB network outperforms existing LSTM-based methods for CPD in fine-grained MFR work mode detection. A new label configuration and a weighted Binary Cross Entropy (wBCE) loss function are utilized for training, which effectively addresses the problem of sparse change point labels.

We reorganized the contribution in the penultimate paragraph of Introduction Section:

“These existing DL methods have achieved automatic identification of multiple work modes with different modulation types under non-ideal conditions, which can be understood as detecting change points of work modes at the modulation type level. Considering that within the same modulation type, there may exist consecutive multiple work modes with different modulation parameters in the MFR pulse sequences. These fine-grained work modes exhibit similar trends under the same modulation type, making it difficult for existing methods to extract discernible features. Therefore, it is necessary to further study CPD algorithms at the modulation parameter level to cope with the rapid and flexible work mode changes of advanced MFRs. There are two problems remain to be solved: first, CPD tasks are often treated as binary classification problems in deep learning, which creates label imbalance issues and makes model fitting difficult. Second, the feature extraction module for detecting fine-grained work mode changes needs to be carefully designed.”

Point 3: In the ‘Dataset Description’, what is the proportion of train set and test set? It seems that the dataset is not enough to support the DL framework.

Response 3:

We apologize for the confusion caused by the “Dataset Description”. In each non-ideal scenario, we train the network through the training dataset containing various levels, and then conduct separate tests at each level by 1000 samples.

Specifically, we reorganized the description in Section 4:

“Taking lost pulse only scenarios as an example, 4000 samples per lost pulse ratio level were generated, i.e. 20000 in total for five levels. The samples are divided into training and test sets with a proportion of 75% and 25%, respectively. Then the model was trained using the 15000 samples in training sets together and separately tested with 1000 samples at each level. The dataset settings were the same for other scenarios. Simulations were also conducted to verify the extent to which the model trained on MNO was affected by the modulation types and the number of change points. 1000 test samples were generated at each level.”

Point 4: In the ‘Simulations and Analysis’, the experimental analysis and visualization results seem insufficient to support the theme proposed in this paper.

Response 4:

In this paper, we solve the problem of MFR work mode change point detection at parameter-level. In order to verify the basic performance of CPD for fine-grained MFR work modes, 1000 test samples with different parameter were tested at each modulation type in Section 4.2. Then, Section 4.3 verified the robustness of our proposed framework under four non-ideal scenarios. Specifically, we reorganized the description of simulation analysis and visualization results in Section 4.3.

As mentioned above, the label configuration and the wBCE loss function also play a significant role in improving performance. We added two simulations to verify their effects. All these simulations support the progressiveness of the proposed framework in fine-grained MFR work mode CPD tasks.

Reviewer 2 Report

Abstract: I propose some results must be highlighted in the abstract to show expressive features of the overall work.

The abbreviation should standardize with a capital letter for each word eg: Change Point Detection (CPD)

A lot of sentences are not comprehensible which may be due to the structure of the sentence. You want to address many issues in one sentence which causes a complex structure and is difficult to understand. Please revise all the sentences and make them simple and easy to understand.

A lot of the issues statements or sentences have no continuation from one to another sentence. For example, a statement in line 27 (CPD) suddenly jumps to some other new topic which is not a continuation of the description of CPD... similarly, a statement in line 32 suddenly jumps to a new issue or topic which has no correlation with the previous line.  By right statement in line 29 should be the first description than line 27. It is better to elaborate on some critical issue or point of the citations so that the paper can reveal a clear crucial point of the research gap.

Overall, rephrasing work for all section 1 (introduction) is considered a major correction.

Author Response

Dear reviewer:

Thank you for your careful reading and the comments concerning our manuscript. Those comments are valuable and very helpful. We have read through comments carefully and have made corrections. Based on your comments, we uploaded the file of the revised manuscript. Revisions in the manuscript are shown using blue highlight for additions, and strikethrough font for deletions. The responses to the reviewer's comments are presented following.

Point 1: Abstract: I propose some results must be highlighted in the abstract to show expressive features of the overall work.

Response 1: Thank you for your suggestion. We have highlighted the effectiveness of the proposed framework in the abstract.

1. The proposed deep learning framework address the challenges for fine-grained MFR work modes under non-ideal conditions.

2. The framework includes a multi-head attention-based bi-directional long short-term memory network to abstract high-order relationships between successive pulses and predict the probability of each pulse being a change point.

3. The framework also addresses the label sparsity problem by utilizing the label configuration and the loss function of training.

4. The simulation results showed that compared with existing methods, the proposed framework effectively improves CPD performance at parameter-level. Moreover, the F1-score was increased by 4.15% under hybrid non-ideal conditions.

Point 2: The abbreviation should standardize with a capital letter for each word eg: Change Point Detection (CPD)

Response 2: Thanks for your careful checks. We are sorry for our carelessness. Based on your comments, we have checked the whole paper repeatedly and standardized the abbreviations.

Point 3: A lot of sentences are not comprehensible which may be due to the structure of the sentence. You want to address many issues in one sentence which causes a complex structure and is difficult to understand. Please revise all the sentences and make them simple and easy to understand.

Response 3: Thanks for your suggestion. We apologize for any confusion caused by the complex structure of our sentences. We tried our best to revise all the sentences in our paper making them easy to understand. The manuscript has been thoroughly revised and edited by a native speaker. We hope it can meet the journal's standard. Thanks so much for your useful comments.

Point 4: A lot of the issues statements or sentences have no continuation from one to another sentence. For example, a statement in line 27 (CPD) suddenly jumps to some other new topic which is not a continuation of the description of CPD... similarly, a statement in line 32 suddenly jumps to a new issue or topic which has no correlation with the previous line.  By right statement in line 29 should be the first description than line 27. It is better to elaborate on some critical issue or point of the citations so that the paper can reveal a clear crucial point of the research gap.

Response 4: We tried our best to improve the manuscript and made some changes to the manuscript. These changes will not influence the content and framework of the paper. 

Specifically, we reorganized the description from line 27 to line 32:

Change Point Detection (CPD) refers to a set of feasible methods for identifying abrupt changes in a time series and can be applied in various fields to extract work mode information [2-7]. The objective of this considered task is to detect all points at which a change in trends occurs, which are defined as change points. In particular, CPD has been proven to be effective in radar signal processing. Early studies for radar work mode CPD were typically based on statistical methods, such as probabilistic analysis [8-9], circular binary segmentation algorithms [10], cumulative sum algorithms [11-13], and manually-designed feature extraction [14]. Nevertheless, heavy reliance on these hypothetical statistical models and manually-designed features usually results in high complexity and poor performance under non-ideal conditions in real applications.

Reviewer 3 Report

The paper is well written and organized to accept for publication after minor revisions. There are some typo errors such as "chang in Figure 4", "(e) A prediected results... in Figure 8" and so on. Please proceed with the overall english correction to help the readers understand. 

Author Response

Dear reviewer:

We would like to thank you for your careful reading, helpful comments, and constructive suggestions, which has significantly improved the presentation of our manuscript. The responses to the reviewer's comments are presented following.

Point 1: The paper is well written and organized to accept for publication after minor revisions. There are some typo errors such as "chang in Figure 4", "(e) A prediected results... in Figure 8" and so on. Please proceed with the overall english correction to help the readers understand.

Response 1:  We thank the reviewer for reading our paper carefully and giving the above positive comments. Based on the reviewer’s suggestions, we have made these corrections and carefully checked the full paper.

Round 2

Reviewer 1 Report

The manuscript can be accepted.